# Glucose Tolerance-Improving Activity of Helichrysoside in Mice and Its Structural Requirements for Promoting Glucose and Lipid Metabolism

**DOI:** 10.3390/ijms20246322

**Published:** 2019-12-14

**Authors:** Toshio Morikawa, Akifumi Nagatomo, Takahiro Oka, Yoshinobu Miki, Norihisa Taira, Megumi Shibano-Kitahara, Yuichiro Hori, Osamu Muraoka, Kiyofumi Ninomiya

**Affiliations:** 1Pharmaceutical Research and Technology Institute, Kindai University, 3-4-1 Kowakae, Higashi-osaka, Osaka 577-8502, Japan; a-nagatomo@jintan.co.jp (A.N.); tmykoka0325@gmail.com (T.O.); sanmokuhoushin@hokuriku.me (Y.M.); Taira_Norihisa@seiwakasei.co.jp (N.T.); kabazakura2@yahoo.co.jp (M.S.-K.); hori.yuichiro.0208@gmail.com (Y.H.); muraoka@phar.kindai.ac.jp (O.M.); ninomiya@phar.kindai.ac.jp (K.N.); 2Antiaging Center, Kindai University, 3-4-1 Kowakae, Higashi-osaka, Osaka 577-8502, Japan

**Keywords:** helichrysoside, acylated flavonol glycoside, glucose tolerance-improving activity, lipid metabolism-promoting activity

## Abstract

An acylated flavonol glycoside, helichrysoside, at a dose of 10 mg/kg/day per os for 14 days, improved the glucose tolerance in mice without affecting the food intake, visceral fat weight, liver weight, and other plasma parameters. In this study, using hepatoblastoma-derived HepG2 cells, helichrysoside, *trans*-tiliroside, and kaempferol 3-*O*-β-d-glucopyranoside enhanced glucose consumption from the medium, but their aglycones and *p*-coumaric acid did not show this activity. In addition, several acylated flavonol glycosides were synthesized to clarify the structural requirements for lipid metabolism using HepG2 cells. The results showed that helichrysoside and related analogs significantly inhibited triglyceride (TG) accumulation in these cells. The inhibition by helichrysoside was more potent than that by other acylated flavonol glycosides, related flavonol glycosides, and organic acids. As for the TG metabolism-promoting activity in high glucose-pretreated HepG2 cells, helichrysoside, related analogs, and their aglycones were found to significantly reduce the TG contents in HepG2 cells. However, the desacyl flavonol glycosides and organic acids derived from the acyl groups did not exhibit an inhibitory impact on the TG contents in HepG2 cells. These results suggest that the existence of the acyl moiety at the 6′′ position in the D-glucopyranosyl part is essential for glucose and lipid metabolism-promoting activities.

## 1. Introduction

Flavonoids are one of the most abundant classes of secondary plant metabolites. Flavonoids are biosynthesized by the shikimate and acetate-malonate pathways and are comprised of compounds that possess a common C_6_-C_3_-C_6_ skeleton, where two aromatic rings (named ring A and B) are linked via a heterocyclic 4*H*-pyrane ring (ring C). Modification of the 15-carbon skeleton through different oxidation levels and substituents to ring C gives rise to different classes of flavonoids, such as flavones, flavonols, flavanones, chalcones, dihydroflavonols (flavanonols), isoflavones, aurones, anthocyanidins, leucoanthocyanidines (flavan-3,4-diols), and flavan-3-ols. They naturally occur in not only aglycone forms, but also as glycosylated and/or acylated derivatives and oligomeric and polymeric structures, such as the flavan-3-ol-derived condensed tannins and proanthocyanidins [1,2,3,4,5]. Flavonoid health benefits are well-recognized, such as their antioxidant properties, properties for weight management, cardiovascular disease protection, anti-allergic activity, vascular fragility, prevention of viral and bacterial infections, anti-inflammatory activity, age-related neurodegenerative disease prevention, anti-platelet aggregation effects, and cancer protection, etc. [2,3,4,5,6,7,8]. Our studies on bioactive constituents from medicinal and/or food resources have reported several bio-functional properties of flavonoids. These included aldose reductase inhibitory [9,10,11,12], anti-platelet aggregation [9], anti-allergic [12,13,14], anti-inflammatory [12,15,16,17,18], aminopeptidase N inhibition [11,17,19], hepatoprotective [20], gastroprotective [21], melanogenesis inhibition [22], and dipeptidyl peptidase-IV inhibitory [23] activities. This paper deals with the practical synthesis and glucose tolerance-improving activity of helichrysoside (**1** = quercetin 3-*O*-(6′′-*O*-*trans*-*p*-coumaroyl)-β-d-glucopyranoside), isolated from *Helichrysum kraussii* and *H. stoechas* [24,25] by other research groups. Furthermore, synthetic studies of the related analogs of 1 (**2**–**15**, Figure 1) were also carried out, as well as characterization of its structural requirements for glucose and lipid metabolism in HepG2 cells. In our previous report, an acylated flavonol glycoside, *trans*-tiliroside (**16** = kaempferol 3-*O*-(6′′-*O*-*trans*-*p*-coumaroyl)-β-d-glucopyranoside), isolated from the fruit of *Rosa canina*, was found to suppress visceral fat weight gain and improve glucose tolerance in mice [26]. The structures of **1** and **16** are similar: the former has a *p*-coumaroyl ester at the 6-position in the β-D-glucopyranosyl moiety of quercetin 3-*O*-β-d-glucopyranoside (isoquercitrin, **17**), while the latter has the common acyl group at the same position of kaempferol 3-*O*-β-d-glucopyranoside (**18**).

## 2. Results and Discussion

### 2.1. Synthesis of Acylated Flavonol Glycosides (***1**–**15***)

Rutin (**19** = quercetin 3-*O*-α-l-rhamnopyraniosyl(1→6)-β-d-glucopyranoside), constructed with quercetin (**20**) as an aglycone, is one of the most widely distributed naturally occurring flavonoids and has been reported to have several pharmacological activities, such as anti-oxidant, anti-inflammatory, anti-diabetic, anti-adipogenic, and neuroprotective effects, and has been used in hormone therapy [27,28,29,30]. In order to achieve the practical synthesis of **1** from **19**, the most inexpensive and commercially available flavonoid, the optimal conditions for enzymatic hydrolysis of the terminal rhamnosyl part were investigated. Therefore, the practical derivation from **19** to **17** was carried out using naringinase (from *Penicillium decumbens*) under an optimal pH and temperature (pH 7 and 50 °C), and the time course of the reaction mixture was monitored by high performance liquid chromatography (HPLC) analysis (Appendix A). As shown in Table 1, the highest content of **17** in the reaction mixture of 2 h was observed. By applying these conditions, a large-scale derivation of **17** (6.50 g and 14.0 mmol, 56.9%) from **19** (15.0 g and 24.6 mmol) was achieved.

Synthesis of the acylated flavonol glycosides, including helichrysoside (**1**–**15**) from **17**, using the corresponding acylation reactions, was carried out. Therefore, protection of a phenol group in *p*-coumaric acid (**22**) with *tert*-butyldiphenylsilyl chloride (TBDPSCl) yielded the corresponding silyl ether, **22a**. Acylation of **17** with **22a** in the presence of 1-ethyl-3-(3-dimethylaminopropyl)carbodiimide hydrochloride (EDC·HCl) and 4-dimethylaminopyridine (4-DMAP) in pyridine, followed by deprotection with tetrabutylammonium fluoride (TBAF), provided **1** with a 36.9% yield (Scheme 1).

In a similar procedure to that of **1** [31,32], compounds **3**, **4**, **5**, **6**, **7** [33], **8** [34], **10** [33], **12** [33,35,36], **14**, and **15** were synthesized with **17** and the corresponding organic acids. The *cis*-isomer of **1**, quercetin 3-*O*-(6′′-*O*-*cis*-*p*-coumaroyl)-β-d-glucopyranoside (**2**), was derived under a UV lamp in a methanol solution of **1**. Using a similar procedure, quercetin 3-*O*-(6′′-*O*-*cis*-caffeoyl)-β-d-glucopyranoside (**9**), quercetin 3-*O*-(6′′-*O*-*cis*-feruloyl)-β-d-glucopyranoside (**11**), and quercetin 3-*O*-(6′′-*O*-*cis*-cinnamoyl)-β-d-glucopyranoside (**13**) were also isomerized from **8**, **10**, and **12**, respectively. Among these synthetic products, known compounds (**1**, **7**, **8**, **10**, and **12**) were identified by a comparison of their physicochemical data with those of authentic samples or with reported values. The structural determination of new compounds (**2**–**6**, **9**, **11**, and **13**–**15**) was elucidated on their spectroscopic properties, including the ^13^C-NMR data, as shown in Appendix A.

### 2.2. Effect of Helichrysoside (***1***) on the Liver Triglyceride (TG) Content and Glucose Tolerance Test after 14 Days of Administration in Mice

Diabetes is characterized by a high incidence of cardiovascular disease and poor control of hyperglycemia caused by insulin resistance (IR). IR can be defined as the inability of insulin to stimulate glucose uptake into the liver, skeletal muscle, or adipose tissue. Hyperglycemia is an important factor contributing to the development of atherosclerosis, and is relevant to the pathophysiology of late diabetic complications. Therefore, improving IR may form part of the strategy for the prevention and management of cardiovascular disease in diabetes [37,38]. We have reported that several anti-diabetogenic therapeutic candidates obtained from natural resources, such as acylated flavonol glycosides from *Sinocrassula indica* [39]; saponins from *Borassus flabellifer* [40]; and thiosugars from *Salacia reticulata*, *S. oblonga*, and *S. chinensis* [41,42,43,44,45,46], showed the inhibition of postprandial hyperglycemia and/or improvement of glucose tolerance in sugar-loaded animal models. As mentioned above, the structure of **16** isolated from *R. canina* [26] is quite similar to **1**, so we presumed that **1** also exhibits similar anti-diabetogenic activity to **16** in an in vivo study. To continue our search for new candidates of the anti-diabetogenic and/or anti-diabetic principles and to evaluate the anti-diabetogenic effect of **1**, the effect of 14 days of the continuous administration of **1** on glucose tolerance was performed in mice. Following this continuous administration, **1** was found to significantly suppress the increase in blood glucose levels at doses of 1 and 10 mg/kg/day per os (p.o.), at 60 min post glucose loading (Figure 2 and Appendix A). The area under the curve (AUC) of blood glucose levels was significantly reduced at the dose of 10 mg/kg/day (p.o.). As indicated in Appendix A, the continuous administration of **1** tended to reduce the weights of visceral fat and the liver and the liver TG content, without affecting the food intake and other plasma parameters, including plasma TG, total cholesterol, and free fatty acids (FFA).

### 2.3. Effects on Glucose Consumption in HepG2 Cells

The liver is one of the tissues important for maintaining blood glucose homeostasis and greatly affects the formation of abnormal glucose tolerance [47]. Since the improving activity of helichrysoside (**1**) in terms of glucose tolerance in mice was observed in the previous section, we investigated the effects of **1** and its related compounds (**16**–**18** and **20**–**22**), to clarify the structural requirement of glucose consumption using human hepatoblastoma-derived HepG2 cells. As shown in Table 2, the glucose concentration in the medium was found to be significantly reduced at 6 days pretreatment with **1**, *trans*-tiliroside (**16**), kaempferol 3-*O*-β-d-glucopyranoside (**18**), and metformin. On the other hand, the desacyl derivative of **1**, quercetin 3-*O*-β-d-glucopyranoside (**17**); the aglycones of **1** and **16**, quercetin (**20**) and kaempferol (**21**); and *trans*-*p*-coumaric acid (**22**) did not result in changes in the glucose concentration in the medium. These results suggested that the *p*-coumaroyl moiety at the 6′′ position in the D-glucopyranosyl part was essential for promoting glucose consumption. Recent related studies have reported that compounds **17**, **20**, and **21** promoted glucose uptake into muscle and hepatocytes [48,49,50]. Due to the long-term treatment of test samples of cells in our study, the treatment with compounds **17**, **20**, and **21** showed cytotoxicity at the concentration of 30–100 µM.

### 2.4. Effects on Lipid Metabolism in HepG2 Cells

A fatty liver is recognized as a significant risk factor for serious liver diseases [51,52]. A strong causal link has been identified between fatty liver diseases and hyperinsulinemia, caused by insulin resistance [53,54]. Therefore, a fatty liver is considered to be closely associated with obesity and type 2 diabetes [54]. In previous studies on the identification of anti-fatty liver principles from natural medicines, several flavonoids [55,56,57,58] were revealed to inhibit lipid accumulation in HepG2 cells. Similarly, we also reported that several megastigmanes [59], diterpenes [60], and limonoids [61] inhibited lipid metabolism in high glucose-pretreated HepG2 cells.

Intracellular TG accumulated in HepG2 cells via increasing the expression of lipogenesis-related proteins, such as sterol regulatory element-binding protein 1c (SREBP-1c) and fatty acid synthase (FAS), when cultured in high glucose-containing medium [55,62]. To characterize this phenomenon, we examined the inhibitory effects of the acylated flavonol glycosides (**1**–**16**) and related compounds (**17**–**22**, **24**, **26**, and **28**–**31**) on (i) high glucose-induced TG accumulation in HepG2 cells and (ii) TG contents in high glucose-pretreated HepG2 cells.

As shown in Table 3, several acylated flavonol glycosides, such as helichrysoside (**1**), quercetin 3-*O*-(6′′-*O*-*cis*-*p*-coumaroyl)-β-d-glucopyranoside (**2**), quercetin 3-*O*-(6′′-*O*-*trans*-*p*-methylcoumaroyl)-β-d-glucopyranoside (**3**), quercetin 3-*O*-(6′′-*O*-trimethylgalloyl)-β-d-glucopyranoside (**15**), and *trans*-tiliroside (**16**), significantly inhibited high glucose-induced TG accumulation in HepG2 cells (% of control at 100 µM: **1** (76.9 ± 2.3%), **2** (80.4 ± 1.2%), **3** (58.2 ± 7.5%), **15** (85.8 ± 4.1%), and **16** (82.3 ± 3.0%)). In contrast, the other acylated flavonol glycosides (**4**–**14**), related flavonol glycosides (**17**–**19**), and organic acids, which related to the corresponding acyl groups (**22**–**31**), did not show significant inhibitory activity up to a concentration of 100 µM. As for the inhibitory effects of the corresponding aglycones, quercetin (**20**, 45.7 ± 0.4% at 100 µM) and kaempferol (**21**, 25.5 ± 1.4% at 100 µM) showed stronger activity than that of the acylated flavonol glycosides, with cytotoxicity under the effective concentrations (data not shown). The structural requirements of the acylated flavonol glycosides were assessed and showed that (1) the acylated flavonol glycosides with a *p*-coumaroyl, *p*-methylcoumaryl, or trimethylgalloyl moiety as the acyl group in the 6′′ position of the D-glucopyranosyl part are essential for the activity, and (2) the glycoside structure contributes to reducing the cytotoxicity.

On the other hand, all the tested acylated flavonol glycosides (**1**–**16**) and their aglycones (**20** and **21**) were found to significantly inhibit the TG content in high glucose-pretreated HepG2 cells at a concentration of 100 µM, as shown in Table 4. Specifically, the 6′′-*O*-acylated quercetin 3-*O*-β-d-glucopyranosides structure, having a *trans*- and *cis*-*p*-coumaroyl (% of control at 10 µM: **1** (82.9 ± 1.3) and **2** (86.0 ± 3.2%)), *trans*-*p*-methylcoumaroyl (**3**, 87.4 ± 0.9%), *trans*-*m*-coumaroyl (**6**, 87.5 ± 4.2%), *trans*-*m*-methylcoumaroyl (**7**, 88.4 ± 2.0%), *trans*- and *cis*-cinnamoyl (**12** (87.2 ± 2.1%) and **13** (88.9 ± 1.8%)), and vanilloyl moiety (**14**, 75.0 ± 7.3%), showed potent activities. However, the corresponding flavonol glycosides (**17**–**19**) and organic acids (**22**, **24**, **26**, and **28**–**31**) lacked this potency. Based on these results, the following structural requirements can be concluded: the acylated flavonol glycosides with a *p*- or *m*-coumaroyl, *p*- or *m*-methylcoumaryl, cinnamoyl, or vanilloyl moiety with the acyl group in the 6′′ position of the D-glucopyranosyl part are essential for the potent inhibition of TG content in high glucose-pretreated HepG2 cells.

Recently, it has been reported that derivatives of **16** activate adenosine 5′-monophosphate-activated protein kinase (AMPK) in 3T3-L1 cells [63] and stimulate glucose transporter (GLUT) 4 translocation in skeletal muscle cells [64]. AMPK is known as a key molecule involved in regulating glucose and lipid metabolism in the liver. From the similarity of the structure, the activities exhibited by **1** and its analogs in this study may have been caused via the same mechanism, but further investigations are needed to clarify the details.

## 3. Materials and Methods

### 3.1. Chemicals and Reagents

Rutin and naringinase were purchased from Sigma-Aldrich Co. LLC., St. Louis, MO, USA. DMF, *tert*-butyldiphenylsilyl chloride (TBDPSCl), EDC·HCl, 4-DMAP, TBAF, and tetrahydrofuran (THF) were purchased from Tokyo Chemical Industry Co., Ltd., Tokyo, Japan. Flavonols (**17**, **18**, and **20**–**21**), organic acids (**22**–**31**), and other chemicals, unless otherwise indicated, were purchased from Nakalai Tesque Inc., Kyoto, Japan.

### 3.2. General Experimental Procedures

The following instruments were used to obtain spectroscopic data: specific rotations, Horiba SEPA-300 digital polarimeter (*l* = 5 cm); UV spectra, Shimadzu UV-1600 spectrometer; IR spectra, Shimadzu FTIR-8100 spectrometer; FAB-MS and high-resolution MS, JEOL JMS-SX 102A mass spectrometer; ESIMS and HRESIMS, Exactive Plus mass spectrometer (Thermo Fisher Scientific Inc., MA, USA); ^1^H-NMR spectra, JEOL JNM-ECA600 (600 MHz) and JNM-ESC400 (400 MHz) spectrometers; ^13^C-NMR spectra, JEOL JNM-ECA600 (150 MHz) and JNM-ESC400 (100 MHz) spectrometers, with tetramethylsilane as an internal standard; HPLC detector, Shimadzu SPD-10A*vp* UV-VIS detectors; and HPLC column, Cosmosil 5C_18_-MS-II (Nacalai Tesque Inc.). The following experimental conditions were used for column chromatography (CC): ordinary-phase silica gel CC, silica gel 60N (Kanto Chemical Co., Tokyo, Japan; 63–210 mesh, spherical, neutral), normal-phase thin-layer chromatography (TLC), pre-coated TLC plates with silica gel 60F_254_ (Merck, Darmstadt, Germany; 0.25 mm), with detection achieved by spraying with 1% Ce(SO_4_)_2_–10% aqueous H_2_SO_4_, and followed by heating.

### 3.3. Enzymatic Hydrolysis of Rutin (***19***) Monitored by HPLC

A suspension of rutin (**19**, 100.0 mg) in H_2_O (50 mL) was mixed and stirred at 50 °C in a water bath for a few minutes. Then, naringinase (5.0 mg) was added to the suspension to start the reaction. Aliquots (1 mL) of the reaction mixture after 0, 5, and 30 min and 1, 1.5, 2, 3, 4.5, 8, and 24 h were transferred into a 10 mL volumetric flask and methanol was added to make up the volume, respectively. Each solution was filtered through a syringe filter (0.45 µm), and an aliquot of 1 µL was subjected to the following HPLC analytical conditions.

A series LC-20A Prominence HPLC system (version 3.40, Shimadzu Co., Kyoto, Japan) was equipped with a UV-VIS detector, a binary pump, a degasser, an autosampler, a thermostatic column compartment, and a control module. The chromatographic separation was performed on a Cosmosil 5C_18_-MS-II (3 µm particle size, 150 × 2.0 mm i.d., Nacalai Tesque Inc., Kyoto, Japan) operated at 40 °C with mobile phase A (acetonitrile) and B (H_2_O containing 0.1% acetic acid). The gradient program was as follows: 0–3 min (A:B = 20:80, v/v) → 10–15 min (90:10, v/v) → 15–25 min (20:80, v/v, hold). The flow rate was 0.2 mL/min with UV detection at 254 nm and the injection volume was 1 µL. The standard curves were prepared with five concentration levels in the range of 25–400 µg/mL (25, 50, 100, 200, and 400 µg/mL, respectively). Linearity for each compound, such as rutin (**19**), quercetin 3-*O*-*β*-d-glucopyranoside (**17**), and quercetin (**20**), was plotted using linear regression of the peak area versus concentration. The coefficient of correlation (*R*^2^) was used to judge the linearity (Appendix A).

### 3.4. Practical Derivation from Rutin (***19***) to Quercetin 3-O-β-d-glucopyranoside (***17***) by Naringinase

Naringinase (750.0 mg) was added to a suspension of rutin (**19**, 15.0 g, 24.6 mmol) in H_2_O (7.5 L), and the mixture was stirred at 50 °C for 2 h. Removal of the solvent from the reaction mixture was carried out under reduced pressure using EtOH as an azeotropic solvent to give a crude product, which, on silica gel CC (500 g, CHCl_3_/MeOH/H_2_O (10:3:0.4, v/v/v)), gave a title compound (**17**, 6.50 g, 56.9%).

### 3.5. Synthesis of Helichrysoside (***1***)

Under an argon atmosphere, imidazole (1.50 g, 22.0 mmol, 3.2 eq) and TBDPSCl (4.54 g, 16.5 mmol, 2.4 eq) were added to a solution of *trans*-*p*-coumaric acid (**22**, 1.30 g, 6.88 mmol) in dry-DMF (12.0 mL), and the mixture was stirred at 40 °C for 16 h. The reaction mixture was poured into ice-water and extracted with EtOAc, before being washed with brine. The extract was condensed under a reduced pressure to give a white solid, which was dissolved in CHCl_3_/MeOH (10:7, v/v, 17 mL) and acidified by 1 M HCl until pH 3.0. After stirring at room temperature for 1.5 h, the reaction mixture was condensed under a reduced pressure to give a pale yellow oil, which was crystallized in *n*-hexane/EtOAc (9:1, v/v, 20 mL) to give **22a** (2.12 g, 76.5%).

4-*O*-*tert*-Butyldiphenylsilyl ether of *trans*-*p*-coumaric acid (**22a**): ^1^H NMR (400 MHz, CDCl_3_): *δ* 1.10 (9H, s, *tert*-Bu), 6.23, 7.65 (1H each, both d, *J* = 16.0 Hz, H-8 and 7), 6.76, 7.30 (2H each, both d, *J* = 8.7 Hz, H-3,5 and 2,6), [7.37 (4H, m), 7.43 (2H, m), 7.70 (4H, dd, *J* = 1.9, 8.2 Hz), arom.]. ^13^C NMR (100 MHz, CDCl_3_): *δ*_C_ 127.2 (C-1), 129.8 (C-2,6), 120.3 (C-3,5), 158.1 (C-4), 146.7 (C-7), 114.8 (C-8), 172.3 (C-9), 132.3, 135.4, 127.9, 130.1 (arom. C-1, 2,6, 3,5, 4), 19.4 (CH_3_*C*-), 26.4 (*C*H_3_C-).

Compound **22a** (0.40 g, 0.96 mmol, 1.2 eq), EDC·HCl (0.31 g, 1.60 mmol, 2.0 eq), and 4- DMAP (0.15 g, 1.20 mmol, 1.5 eq) were added to a solution of **17** (0.37 g, 0.80 mmol) in pyridine (4.0 mL), and the mixture was stirred at 50 °C for 12 h. The reaction mixture was condensed under a reduced pressure to give a crude product. Then, a solution of the crude product in THF (5.0 mL) was added to TBAF (*ca.* 1.0 mol/L in THF, 800 µL, 0.80 mmol, 1.0 eq) at room temperature for 1 h. The reaction mixture was quenched in H_2_O, and removal of the solvent under a reduced pressure then furnished a residue, which, on silica gel CC (10 g, CHCl_3_/MeOH/H_2_O (10:3:0.4, v/v/v)), gave a title compound (**1**, 0.18 g, 36.9%).

### 3.6. Synthesis of ***3**–**8**, **10**, **12**, **14***, and ***15***

In a manner similar to that used for **22a** (*vide supra*), *trans*-*o*-coumaric acid (**24**), *trans*-*m*-coumaric acid (**26**), *trans*-caffeic acid (**28**), *trans*-ferulic acid (**29**), and vanillic acid (**31**) were derived to yield the corresponding silyl ether derivatives, **24a** (95.1%), **26a** (94.6%), **28a** (393.0%), **29a** (87.3%), and **31a** (98.5%), respectively.

2-*O*-*tert*-Butyldiphenylsilyl ether of *trans*-*o*-coumaric acid (**24a**): ^1^H NMR (400 MHz, CDCl_3_): *δ* 1.15 (9H, s, *tert*-Bu), 6.49, 8.53 (1H each, both d, *J* = 16.0 Hz, H-8 and 7), 6.48 (1H, dd, *J* = 1.8, 8.7 Hz, H-3), 6.88 (1H, br dd, *J* = ca. 8, 8 Hz, H-5), 6.96 (1H, ddd, *J* = 1.8, 8.3, 8.7 Hz, H-4), 7.59 (1H, dd, *J* = 1.8, 7.8 Hz, H-6), (7.39 (4H, m), 7.44 (2H, m), 7.72 (4H, dd, *J* = 1.8, 8.3 Hz), arom.). ^13^C NMR (100 MHz, CDCl_3_): *δ*_C_ 125.0 (C-1), 154.6 (C-2), 119.9 (C-3), 131.5 (C-4), 121.4 (C-5), 127.5 (C-6), 142.3 (C-7), 117.1 (C-8), 172.6 (C-9), 132.2, 135.4, 127.9, 130.1 (arom. C-1, 2,6, 3,5, 4), 19.6 (CH_3_*C*-), 26.5 (*C*H_3_C-).

3-*O*-*tert*-Butyldiphenylsilyl ether of *trans*-*m*-coumaric acid (**26a**): ^1^H NMR (400 MHz, CDCl_3_): *δ* 1.12 (9H, s, *tert*-Bu), 6.17, 7.59 (1H each, both d, *J* = 16.0 Hz, H-8 and 7), 6.78 (1H, br d, *J* = ca. 8 Hz, H-4), 6.94 (1H, br s, H-2), 7.04 (br d, *J* = ca. 8 Hz, H-6), 7.10 (dd, *J* = 7.8, 7.8 Hz, H-5), (7.38 (4H, m), 7.44 (2H, m), 7.71 (4H, m), arom.). ^13^C NMR (100 MHz, CDCl_3_): *δ*_C_ 135.2 (C-1), 119.1 (C-2), 156.0 (C-3), 122.2 (C-4), 129.7 (C-5), 121.5 (C-6), 146.8 (C-7), 117.3 (C-8), 172.1 (C-9), 132.2, 135.5, 127.9, 130.1 (arom. C-1, 2,6, 3,5, 4), 19.5 (CH_3_*C*-), 26.5 (*C*H_3_C-).

3,4-Di-*O*-*tert*-butyldiphenylsilyl ether of *trans*-caffeic acid (**28a**): ^1^H NMR (400 MHz, CDCl_3_): *δ* 1.14, 1.18 (9H each, both s, *tert*-Bu), 5.48, 7.17 (1H each, both d, *J* = 16.0 Hz, H-8 and 7), 6.39 (1H, d, *J* = 8.2 Hz, H-5), 6.53 (1H, dd, *J* = 1.8, 8.2 Hz, H-6), 6.59 (1H, d, *J* = 1.8 Hz, H-2), (7.40 (8H, m), 7.45 (4H, m), 7.79 (8H, m), arom.). ^13^C NMR (100 MHz, CDCl_3_): *δ*_C_ 126.8 (C-1), 119.4 (C-2), 146.5 (C-3), 148.9 (C-4), 120.5 (C-5), 122.3 (C-6), 146.6 (C-7), 114.4 (C-8), 172.0 (C-9), 132.5/132.9, 135.4/135.6, 127.9/128.0, 130.0/130.2 (arom. C-1, 2,6, 3,5, 4), 19.5, 19.5 (CH_3_*C*-), 26.6, 26.8 (*C*H_3_C-).

4-*O*-*tert*-Butyldiphenylsilyl ether of *trans*-ferulic acid (**29a**): ^1^H NMR (400 MHz, CDCl_3_): *δ* 1.11 (9H, s, *tert*-Bu), 3.60 (3H, s, -OCH_3_), 6.23, 7.64 (1H each, both d, *J* = 16.0 Hz, H-8 and 7), 6.69 (1H, d, *J* = 8.2 Hz, H-5), 6.86 (1H, dd, *J* = 1.8, 8.2 Hz, H-6), 6.95 (1H, d, *J* = 1.8 Hz, H-2), (7.35 (4H, m), 7.41 (2H, m), 7.70 (4H, dd, *J* = 1.8, 7.8 Hz), arom.). ^13^C NMR (100 MHz, CDCl_3_): *δ*_C_ 127.7 (C-1), 111.2 (C-2), 150.8 (C-3), 147.9 (C-4), 120.4 (C-5), 122.5 (C-6), 147.1 (C-7), 114.7 (C-8), 172.3 (C-9), 133.1, 135.3, 127.6, 129.8 (arom. C-1, 2,6, 3,5, 4), 19.8 (CH_3_*C*-), 26.6 (*C*H_3_C-).

3-*O*-*tert*-Butyldiphenylsilyl ether of vanillic acid (**31a**): ^1^H NMR (400 MHz, CDCl_3_): *δ* 1.12 (9H, s, *tert*-Bu), 3.61 (3H, s, -OCH_3_), 6.37 (1H, d, *J* = 9.2 Hz, H-5), 7.47 (1H, dd, *J* = 1.8, 9.2 Hz, H-6), 7.48 (1H, d, *J* = 1.8 Hz, H-2), (7.35 (4H, m), 7.41 (2H, m), 7.69 (4H, m), arom.). ^13^C NMR (100 MHz, CDCl_3_): *δ*_C_ 122.4 (C-1), 113.5 (C-2), 150.4 (C-3), 150.3 (C-4), 119.8 (C-5), 124.0 (C-6), 171.7 (C-9), 132.9, 135.2, 127.6, 129.8 (arom. C-1, 2,6, 3,5, 4), 19.8 (CH_3_C-), 26.5 (CH_3_C-).

In a manner similar to that used for the preparation of **1** (*vide supra*), dehydration condensation of **17** (500.0 mg, 1.08 mmol) and **24a**, **26a**, **28a**, **30a**, and **31a** (521.1, 521.1, 521.1, 849.6, 558.6, and 526.3 mg, respectively, 1.29 mmol, 1.2 eq) were conducted to yield the corresponding 6′′-*O*-acylated quercetin 3-β-d-glucopyranosides, **4** (134.6 mg, 31.0%), **6** (147.7 mg, 34.0%), **8** (248.5 mg, 35.1%), **10** (176.9 mg, 38.0%), and **14** (130.7 mg, 29.8%), respectively. Syntheses of the 6′′-*O*-acylated quercetin 3-*β*-d-glucopyranosides, **3** (39.0 mg, 20.3%), **5** (63.2 mg, 32.9%), **7** (57.2 mg, 29.8%), **12** (53.2 mg, 33.3%), and **15** (75.2 mg, 32.9%) were also carried out by the dehydration condensation of **17** with *trans*-*p*-methylcoumaric acid (**23**), *trans*-*o*-methylcoumaric acid (**25**), *trans*-*m*-methylcoumaric acid (**27**), *trans*-cinnamic acid (**30**), and trimethylgallic acid, respectively, without the deprotection procedure of the *tert*-butyldiphenylsilyl (TBDPS) ether group.

Quercetin 3-*O*-(6′′-*O*-*trans*-*p*-methylcoumaroyl)-β-d-glucopyranoside (**3**): A yellow powder, high-resolution positive-ion FABMS: Calcd for C_31_H_28_O_14_Na (M+Na)^+^: 647.1377. Found: 647.1373. ^1^H NMR (600 MHz, DMSO-*d*_6_): *δ* 3.21 (1H, dd, *J* = 8.9, 9.0 Hz, H-4′′), 3.29 (1H, dd, *J* = 8.8, 9.0 Hz, H-3′′), 3.31 (1H, dd, *J* = 7.4, 8.8 Hz, H-2′′), 3.40 (1H, ddd, *J* = 2.0, 6.9, 8.9 Hz, H-5′′), 3.82 (3H, s, -OCH_3_), (4.30 (1H, dd, *J* = 2.0, 11.8 Hz), 4.66 (1H, dd, *J* = 6.9, 11.8 Hz), H_2_-6′′), 5.49 (1H, d, *J* = 7.4 Hz, H-1′′), 6.13, 6.33 (1H each, both d, *J* = 2.0 Hz, H-6 and 8), 6.17, 7.36 (1H each, both d, *J* = 16.0 Hz, H-8′′′ and 7′′′), 6.83 (1H, d, *J* = 8.4 Hz, H-5′), 6.95, 7.44 (2H each, both d, *J* = 8.8 Hz, H-3′′′,5′′′ and 2′′′,6′′′), 7.53 (1H, dd, *J* = 2.2, 8.4 Hz, H-6′), 7.55 (1H, d, *J* = 2.2 Hz, H-2′), 12.60 (1H, br s, 5-OH). ^13^C NMR (150 MHz, DMSO-*d*_6_): *δ*_C_ given in Appendix A.

Quercetin 3-*O*-(6′′-*O*-*trans*-*o*-coumaroyl)-β-d-glucopyranoside (**4**): A yellow powder, high-resolution positive-ion FABMS: Calcd for C_30_H_26_O_14_Na (M+Na)^+^: 633.1220. Found: 633.1226. ^1^H NMR (600 MHz, DMSO-*d*_6_): *δ* 3.22 (1H, dd, *J* = 8.8, 9.5 Hz, H-4′′), 3.28 (1H, dd, *J* = 8.8, 8.8 Hz, H-3′′), 3.30 (1H, dd, *J* = 7.3, 8.8 Hz, H-2′′), 3.40 (1H, ddd, *J* = 2.2, 6.4, 9.5 Hz, H-5′′), (4.07 (1H, dd, *J* = 6.4, 11.9 Hz), 4.29 (1H, dd, *J* = 2.2, 11.9 Hz), H_2_-6′′), 5.47 (1H, d, *J* = 7.3 Hz, H-1′′), 6.15, 6.38 (1H each, both d, *J* = 2.0 Hz, H-6 and 8), 6.34, 7.73 (1H each, both d, *J* = 16.1 Hz, H-8′′′, 7′′′), 6.83 (1H, ddd, *J* = 1.7, 7.8, 8.7 Hz, H-5′′′), 6.84 (1H, d, *J* = 8.2 Hz, H-5′′), 6.93 (1H, dd, *J* = 1.7, 8.3 Hz, H-3′′′), 7.22 (1H, ddd, *J* = 1.6, 8.3, 8.7 Hz, H-4′′′), 7.38 (1H, dd, *J* = 1.6, 7.8 Hz, H-6′′′), 7.52 (1H, dd, *J* = 2.0, 8.2 Hz, H-6′′), 7.55 (1H, d, *J* = 2.0 Hz, H-2′′), 12.58 (1H, br s, 5-OH). ^13^C NMR (150 MHz, DMSO-*d*_6_): *δ*_C_ given in Appendix A.

Quercetin 3-*O*-(6′′-*O*-*trans*-*o*-methylcoumaroyl)-β-d-glucopyranoside (**5**): A yellow powder, high-resolution positive-ion FABMS: Calcd for C_31_H_28_O_14_Na (M+Na)^+^: 647.1377. Found: 647.1381. ^1^H NMR (600 MHz, DMSO-*d*_6_): *δ* 3.21 (1H, m, H-4′′), 3.33 (2H, m, H-2′′, 3′′), 3.41 (1H, ddd, *J* = 2.3, 6.9, 9.5 Hz, H-5′′), 3.82 (3H, s, -OCH_3_), (4.11 (1H, dd, *J* = 6.9, 12.1 Hz), 4.30 (1H, dd, *J* = 2.3, 12.1 Hz), H_2_-6′′), 5.48 (1H, d, *J* = 7.5 Hz, H-1′′), 6.07, 6.30 (1H each, both d, *J*= 2.0 Hz, H-6 and 8), 6.32, 7.70 (1H each, both d, *J* = 16.1 Hz, H-8′′′ and 7′′′), 6.83 (1H, d, *J* = 8.9 Hz, H-5′′), 6.98 (1H, br dd, *J* = ca. 8, 8 Hz, H-5′′′), 7.05 (1H, br d, *J* = ca. 8 Hz, H-3′′′), 7.41 (1H, ddd, *J* = 1.7, 8.2, 8.3 Hz, H-4′′′), 7.47 (1H, dd, *J* = 1.7, 8.2 Hz, H-6′′′), 7.52 (1H, dd, *J* = 2.3, 8.9 Hz, H-6′′), 7.53 (1H, d, *J* = 2.3 Hz, H-2′′), 12.57 (1H, br s, 5-OH). ^13^C NMR (150 MHz, DMSO-*d*_6_): *δ*_C_ given in Appendix A.

Quercetin 3-*O*-(6′′-*O*-*trans*-*m*-coumaroyl)-β-d-glucopyranoside (**6**): A yellow powder, high-resolution positive-ion FABMS: Calcd for C_30_H_26_O_14_Na (M+Na)^+^: 633.1220. Found: 633.1226. ^1^H NMR (600 MHz, DMSO-*d*_6_): *δ* 3.22 (1H, dd, *J* = 8.8, 9.0 Hz, H-4′′), 3.28 (1H, dd, *J* = 7.6, 8.9 Hz, H-2′′), 3.31 (1H, dd, *J* = 8.8, 8.9 Hz, H-3′′), 3.40 (1H, ddd, *J* = 2.0, 6.6, 9.0 Hz, H-5′′), (4.08 (1H, dd, *J* = 6.6, 11.9 Hz), 4.30 (1H, dd, *J* = 2.0, 11.9 Hz), H_2_-6′′), 5.46 (1H, d, *J* = 7.6 Hz, H-1′′), 6.12, 6.33 (1H each, both br s, H-6, 8), 6.24, 7.35 (1H each, both d, *J* = 16.0 Hz, H-8′′′ and 7′′′), 6.82 (1H, d, *J* = 8.4 Hz, H-5′), 6.83 (1H, br d, *J* = ca. 8 Hz, H-4′′′), 6.91 (1H, br d, *J* = ca. 8 Hz, H-6′′′), 6.94 (1H, br s, H-2′′′), 7.20 (1H, dd, *J* = 8.0, 8.0 Hz, H-5′′′), 7.52 (1H, dd, *J* = 2.2, 8.4 Hz, H-6′), 7.54 (1H, d, *J* = 2.2 Hz, H-2′), 12.57 (1H, br s, 5-OH). ^13^C NMR (150 MHz, DMSO-*d*_6_): *δ*_C_ given in Appendix A.

Quercetin 3-*O*-(6′′-*O*-vanilloyl)-β-d-glucopyranoside (**14**): A yellow powder, high-resolution positive-ion FABMS: Calcd for C_29_H_26_O_15_Na (M+Na)^+^: 637.1169. Found: 637.1174. ^1^H NMR (600 MHz, DMSO-*d*_6_): *δ* 3.24 (1H, dd, *J* = 8.9, 9.1 Hz, H-4′′), 3.30 (1H, dd, *J* = 8.8, 8.9 Hz, H-3′′), 3.33 (1H, dd, *J* = 7.4, 8.8 Hz, H-2′′), 3.45 (1H, ddd, *J* = 2.1, 6.9, 9.1 Hz, H-5′′), 3.70 (3H, s, -OCH_3_), (4.13 (1H, dd, *J* = 6.9, 11.9 Hz), 4.40 (1H, dd, *J* = 2.1, 11.9 Hz), H_2_-6′′), 5.54 (1H, d, *J* = 7.4 Hz, H-1′′), 6.18, 6.35 (1H each, both d, *J* = 1.9 Hz, H-6 and 8), 6.70 (1H, d, *J* = 8.2 Hz, H-5′′′), 6.78 (1H, d, *J* = 8.4 Hz, H-5′), 7.19 (1H, dd, *J* = 2.0, 8.2 Hz, H-6′′′), 7.27 (1H, d, *J* = 2.0 Hz, H-2′′′), 7.51 (1H, d, *J* = 2.2 Hz, H-2′), 7.53 (1H, dd, *J* = 2.2, 8.4 Hz, H-6′), 12.58 (1H, br s, 5-OH). ^13^C NMR (150 MHz, DMSO-*d*_6_): *δ*_C_ given in Appendix A.

Quercetin 3-*O*-(6′′-*O*-trimethylgalloyl)-β-d-glucopyranoside (**15**): A yellow powder, high-resolution positive-ion FABMS: Calcd for C_31_H_30_O_16_Na (M+Na)^+^: 681.1432. Found: 681.1436. UV (MeOH, nm (log *ε*)): 259 (3.99), 295 (3.62), 359 (3.77). IR (KBr): 3590, 1701, 1655, 1597, 1503, 1341, 1202, 1075 cm^-1^. ^1^H NMR (600 MHz, DMSO-*d*_6_): *δ* 3.23 (1H, dd, *J* = 9.1, 9.7 Hz, H-4′′), 3.30 (2H, m, H-2′′, 3′′), 3.50 (1H, ddd, *J* = 2.4, 7.3, 9.7 Hz, H-5′′), (3.67 (6H, s), 3.73 (3H, s), -OCH_3_), (4.26 (1H, dd, *J* = 7.3, 11.9 Hz), 4.43 (1H, dd, *J* = 2.4, 11.9 Hz), H_2_-6′′), 5.49 (1H, d, *J* = 7.6 Hz, H-1′′), 6.10, 6.30 (1H each, both d, *J* = 2.1 Hz, H-6 and 8), 6.74 (1H, d, *J* = 8.6 Hz, H-5′), 7.48 (1H, br s, H-2′), 7.49 (1H, br d, *J* = ca. 9 Hz, H-6′), 12.48 (1H br s, 5-OH). ^13^C NMR (150 MHz, DMSO-*d*_6_): *δ*_C_ given in Appendix A.

### 3.7. Isomerization of ***1**, **8**, **10***, and ***12***

A methanol solution (20 mL) of **1** (100.0 mg, 0.17 mmol) in a Pyrex tube was left standing for 8 h under a UV lamp (short wave) at room temperature. The reaction mixture was condensed under a reduced pressure to give a crude product, which, on HPLC (Cosmosil 5C_18_-MS-II, MeOH–1% aqueous AcOH (55:45, v/v)), gave the *cis*-isomer **2** (25.1 mg) and a recovering compound (**1**, 69.8 mg). Using the similar procedure, a methanol solution (10 mL) of **8**, **10**, or **12** (each 50.0 mg) was isomerized to the corresponding *cis*-isomer **9** (12.2 mg (recovered **8**, 25.8 mg)), **11** (10.8 mg (recovered **10**, 26.9 mg)), or **13** (14.2 mg (recovered **12**, 30.1 mg)).

Quercetin 3-*O*-(6′′-*O*-*cis*-*p*-coumaroyl)-β-d-glucopyranoside (**2**): A yellow powder, high-resolution positive-ion FABMS: Calcd for C_30_H_26_O_14_Na (M+Na)^+^: 633.1220. Found: 633.1226. ^1^H NMR (600 MHz, CD_3_OD): *δ* 3.29 (1H, dd, *J* = 8.9, 9.0 Hz, H-4′′), 3.42 (2H, m, H-3′′, 5′′), 3.49 (1H, dd, *J* = 7.3, 8.8 Hz, H-2′′), 4.14–4.22 (2H, m, H_2_-6′′), 5.19 (1H, d, *J* = 7.3 Hz, H-1′′), 5.51, 6.69 (1H each, both d, *J* = 12.8 Hz, H-8′′′ and 7′′′), 6.18, 6.30 (1H each, both d, *J* = 1.8 Hz, H-6 and 8), 6.67, 7.49 (2H each, both d, *J* = 8.6 Hz, H-3′′′,5′′′ and 2′′′,6′′′), 6.80 (1H, d, *J* = 8.7 Hz, H-5′), 7.56 (2H, m, H-2′, 6′). ^13^C NMR (150 MHz, CD_3_OD): *δ*_C_ given in Appendix A.

Quercetin 3-*O*-(6′′-*O*-*cis*-caffeoyl)-β-d-glucopyranoside (**9**): A yellow powder, high-resolution positive-ion FABMS: Calcd for C_30_H_26_O_15_Na (M+Na)^+^: 649.1169. Found: 649.1170. ^1^H NMR (600 MHz, DMSO-*d*_6_): *δ* 3.18–3.36 (4H, m, H-2′′–5′′), 3.71(3H, s, -OCH_3_), (4.04 (1H, dd, *J* = 6.4, 11.9 Hz), 4.19 (1H, dd, *J* = 2.2, 11.9 Hz), H_2_-6′′), 5.42 (1H, d, *J* = 7.5 Hz, H-1′′), 5.42, 6.58 (1H each, both d, *J* = 12.8 Hz, H-8′′′ and 7′′′), 6.20, 6.36 (1H each, both d, *J* = 1.7 Hz, H-6 and 8), 6.28 (1H, d, *J* = 2.0 Hz, H-2′′′), 6.67 (1H, d, *J* = 8.5 Hz, H-5′′′), 6.82 (1H, d, *J* = 8.6 Hz, H-5′), 6.98 (1H, dd, *J* = 2.0, 8.5 Hz, H-6′′′), 7.52 (2H, m, H-2′, 6′), 12.59 (1H, br s, 5-OH). ^13^C NMR (150 MHz, DMSO-*d*_6_): *δ*_C_ given in Appendix A.

Quercetin 3-*O*-(6′′-*O*-*cis*-feruloyl)-β-d-glucopyranoside (**11**): A yellow powder, high-resolution positive-ion FABMS: Calcd for C_31_H_28_O_15_Na (M+Na)^+^: 663.1326. Found: 663.1330. ^1^H NMR (600 MHz, DMSO-*d*_6_): *δ* 3.19 (1H, m, H-4′′), 3.27 (2H, m, H-2′′, 3′′), 3.36 (1H, ddd, *J* = 2.3, 6.0, 9.6 Hz, H-5′′), 3.71 (3H, s, -OCH_3_), (4.18 (1H, dd, *J* = 6.4, 11.9 Hz), 4.19 (1H, dd, *J* = 1.8, 11.9 Hz), H_2_-6′′), 5.46 (1H, d, *J* = 7.4 Hz, H-1′′), 5.48, 6.65 (1H each, both d, *J* = 12.8 Hz, H-8′′′ and 7′′′), 6.19, 6.31 (1H each, both d, *J* = 1.8 Hz, H-6 and 8), 6.72 (1H, d, *J* = 8.2 Hz, H-5′′′), 6.81 (1H, d, *J* = 8.7 Hz, H-5′), 7.08 (1H, dd, *J* = 1.8, 8.2 Hz, H-6′′′), 7.45 (1H, br d, *J* = ca. 9 Hz, H-6′), 7.52 (1H, br s, H-2′), 7.59 (1H, d, *J* = 1.8 Hz, H-2′′′), 12.57 (1H, br s, 5-OH). ^13^C NMR (150 MHz, DMSO-*d*_6_): *δ*_C_ given in Appendix A.

Quercetin 3-*O*-(6′′-*O*-*cis*-cinnamoyl)-β-d-glucopyranoside (**13**): A yellow powder, high-resolution positive-ion FABMS: Calcd for C_30_H_26_O_13_Na (M+Na)^+^: 617.1271. Found: 617.1277. ^1^H NMR (600 MHz, DMSO-*d*_6_): *δ* 3.14 (1H, dd, *J* = 8.6, 9.5 Hz, H-4′′), 3.25 (1H, dd, *J* = 7.5, 8.9 Hz, H-2′′), 3.28 (1H, dd, *J* = 8.6, 8.9 Hz, H-3′′), 3.33 (1H, ddd, *J* = 2.3, 6.6, 9.5 Hz, H-5′′), (4.07 (1H, dd, *J* = 6.6, 11.8 Hz), 4.18 (1H, dd, *J* = 2.3, 11.8 Hz), H_2_-6′′), 5.69, 6.81 (1H each, both d, *J* = 12.9 Hz, H-8′′′ and 7′′′), 6.19, 6.33 (1H each, both d, *J* = 2.0 Hz, H-6 and 8), 6.82 (1H, d, *J* = 8.9 Hz, H-5′), 7.27 (4H, m, H-2′′′,6′′′, 3′′′,5′′′), 7.43 (1H, m, H-4′′′), 7.51 (1H, dd, *J* = 2.0, 8.9 Hz, H-6′), 7.51 (1H, d, *J* = 2.0 Hz, H-2′), 12.56 (1H, br s, 5-OH). ^13^C NMR (150 MHz, DMSO-*d*_6_): *δ*_C_ given in Appendix A.

### 3.8. Animals

Male ddY mice were purchased from Kiwa Laboratory Animal Co., Ltd., (Wakayama, Japan). The animals were housed at a constant temperature of 23 ± 2 °C and fed a standard laboratory chow (MF, Oriental Yeast Co., Ltd., Tokyo, Japan). All experiments were performed with conscious mice, unless otherwise noted. The experimental protocol was approved by Kindai University’s Committee for the Care and Use of Laboratory Animals (KAPR-26-004, 1 April 2014).

### 3.9. Effects on the Glucose Tolerance Test in Mice

Effects on the glucose tolerance test after 14 days of administration of **1** in mice were determined according to the previously described protocol [26]. A test sample was administrated orally to male ddY mice (11 weeks old and fed a standard laboratory chow) once a day (10:00–12:00) for 14 days. Body weight and food intake were measured every day before administration of the test sample. Fasting for 20 h was carried out after the final administration, and 10% (w/v) glucose solution was intraperitoneally (i.p.) administrated to mice at 10 mL/kg. Blood samples (*ca.* 0.2 mL) were collected in tubes containing 10 units of heparin sodium from the infraorbital venous plexus before (0 h) and 0.5, 1, and 2 h after the loading of glucose. Mice were then killed by cervical dislocation, and the epididymal, mesenteric, and paranephric fat pads were removed and weighed. Plasma glucose, TG, total cholesterol, and FFA levels were determined using commercial kits (Glucose CII-test Wako, Triglyceride E-test Wako, Cholesterol E-test Wako, and NEFA C-test Wako, respectively, FUJIFILM Wako Pure Chemical Corporation, Tokyo, Japan). After removing the liver, ca. 300 mg of liver tissue was cut and homogenized with 9 mL of distilled water. An aliquot of the homogenate (500 µL) was diluted with distilled water (1 mL) and the TG concentration in the suspension was determined using Triglyceride E-test Wako.

### 3.10. Cell Culture

HepG2 cells (RCB1648, Riken Cell Bank, Tsukuba, Japan) were maintained in Minimum Essential Medium Eagle (MEM, Sigma-Aldrich Co. LLC., St. Louis, MO, USA) containing 10% fetal bovine serum, 1% MEM non-essential amino acids (FUJIFILM Wako Pure Chemical Corporation, Tokyo, Japan), penicillin G (100 units/mL), and streptomycin (100 µg/mL) at 37 °C under 5% CO_2_ atmosphere.

### 3.11. Effects on Glucose Consumption in HepG2 Cells

HepG2 cells were inoculated in a 48-well tissue culture plate (10^5^ cells/well in 150 µL/well in MEM). After 20 h, the medium was replaced with 150 µL/well of Dulbecco’s Modified Eagle’s Medium (DMEM) containing low-glucose (1000 mg/L) and a test sample. Cells were cultured for 6 days, and the medium was replaced every 2 days. The medium was then transferred to 200 µL/well of DMEM containing high-glucose (4500 mg/L) and the cells were cultured. After 20 h, the glucose content in the medium was determined using commercial kits (Glucose CII-test Wako, FUJIFILM Wako Pure Chemical Corporation, Tokyo, Japan). Medium was removed, and the cells were homogenized in distilled water (105 µL/well) by sonication. The protein content in the homogenate was determined using the BCA protein Assay Kit (FUJIFILM Wako Pure Chemical Corporation, Tokyo, Japan). Each test compound was dissolved in DMSO and added to the medium (final DMSO concentration was 0.5%). An anti-diabetic agent, metformin, was used as a reference compound.

### 3.12. Effects on High Glucose-Induced TG Accumulation in HepG2 Cells

HepG2 cells were inoculated in a 48-well tissue culture plate (10^5^ cells/well in 150 µL/well in MEM). After 20 h, the medium was replaced with 150 µL/well of DMEM containing high-glucose and a test sample, which was cultured for 4 days, with medium containing a test sample being replaced every 2 days. Medium was then removed, and the cells were homogenized in distilled water (105 µL/well) by sonication. The TG and protein content in the homogenate were determined using commercial kits (Triglyceride E-test Wako and BCA protein Assay Kit, respectively, FUJIFILM Wako Pure Chemical Corporation, Tokyo, Japan). Data were expressed as the % of control of TG/protein (µg/mg). Each test compound was dissolved in DMSO and was added to the medium (final DMSO concentration was 0.5%). An anti-diabetic agent, metformin, was used as a reference compound.

### 3.13. Effects on TG contents in High Glucose-Pretreated HepG2 Cells

Effects on TG metabolism-promoting activity in high glucose-pretreated HepG2 cells were evaluated according to the method described previously [61], with slight modifications. HepG2 cells were inoculated in a 48-well tissue culture plate (10^5^ cells/well in 150 µL/well in MEM). After 20 h, the medium was replaced with 150 µL/well of DMEM containing high-glucose and cultured for 6 days, with the medium being replaced every 2 days. After accumulation of the lipid, the medium was transferred to 150 µL/well of DMEM containing low-glucose and a test sample, and the cells were cultured. After 20 h, the TG and protein content in the cells were determined by the same manner as described above. Data were expressed as the % of control of TG/protein (µg/mg). Each test compound was dissolved in DMSO and added to the medium (final DMSO concentration was 0.5%). An anti-diabetic agent, metformin, was used as a reference compound.

### 3.14. Statistics

Values are expressed as means ± S.E. One-way analysis of variance (ANOVA) followed by Dunnett’s test was used for statistical analysis. Probability (*p*) values of less than 0.05 were considered significant.

## 4. Conclusions

The present study demonstrated that helichrysoside (**1**), an acylated flavonol glycoside, improved glucose tolerance in ddY mice. In the study, using HepG2 cells, helichrysoside (**1**) was shown to significantly enhance glucose consumption in the medium, inhibit high glucose-induced TG accumulation in cells, and promote the effect of TG metabolism in high glucose-pretreated cells. The results from various acylated flavonol glycosides, flavonol glycosides, flavonols, and organic acids indicated that the acyl group at the 6′′ position in the D-glucopyranosyl part was essential for the improved glucose tolerance activities. Previous evidence, along with this study, suggests that helichrysoside (**1**) might be considered as a possible candidate for the prevention of glucose and lipid metabolism-related disorders.

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
