# Peer review of "Glucose Tolerance-Improving Activity of Helichrysoside in Mice and Its Structural Requirements for Promoting Glucose and Lipid Metabolism"

_ijms, 2019, doi:10.3390/ijms20246322_

Round 1
Reviewer 1 Report
The present paper deals with glucose tolerance-improving activity of helichrysoside, an acylated flavonoid isolated from Helichrysum kraussii
and H. stoechas. The manuscript is well written and quite easy to read.
However, the major problem of the work is its unusual, maybe even wrong concept. Instead of first testing the compound of interest in vitro, the authors first tested it in mice, then in vitro (glucose consumption by HepG2 cells) together with a few commercially available derivatives and finally with a series of in house produced derivatives again in vitro, but testing accumulation of triacylglycerols in the cells.
The rationale of this concept, but also of the mere idea of testing this particular flavonoid must be clearly explained in the manuscript.
For other comments and suggestion in the context of the text, please see the attached quoted manuscript.

Reviewer 2 Report
The major concern of this study is that it lacks the focus. The authors should decide whether they report bioactivity of helichrysoside (main title) or are reporting flavonoid glycosides' profile of synthetic flavodoid glycosides (abstract) or are willing to reveal what is the role of a compound(s) in biological response of hepatocytes. The paper is too confusing to be readable and I strongly suggest the authors to work on text flow and interconnection of study units.
Specific remarks:
Main title: add "in Mice" at the end of the title.
Abstract: it is meaningless to use compounds' labels in Abstract section. Please use full compound names or compound groups.
L18: "improved" instead of "improving".
L19: label "ddY mice" meaningless since is not in use further in the Abstract.
L23: replace "show" with "initiate" or similar.
L23-24: starting with "to clarify" - I'm not sure what the authors meant to say. Please rephrase.
L27: add "these" before "cells".
L39: "structural requirement" as a keyword has broad meaning, hence should be excluded or substituted.
L52-53: the sentence is too trivial. Please rephrase.
L59-64: great job but instead of this self-advertising authors might cite a review article that summarizes flavodoids' bioactivities.
L86: replace "was" with "were".
L102-111: only compound labels should be retained with the reference to a specific table which lists all the compounds. What the cited references are supposed to refer?
L121: please add: "for compound names please refer to..." in the end of the table caption.
L124: I see no connection between this and the previous chapter.
L133-138: wrong syntax.
L141: "Following, this continuous administration..." - wrongly formatted.
L166: replace "of compounds" with "with compounds".
L169: what "S.E.M." means? Standard error? Shouldn't be "S.E."?
L170-171: information about chemicals' purchase should stand in the M&M section, not here. The same applies for Tables 5 and 6.
L177-181: I am quite sure there are studies other than the authors' own that report the same topic.
L199: you cannot start a sentence with "Whereas". Put a comma here instead of full stop and continue with small caption.
L215: delete "of".
L226: here you should discuss this finding in view of the up-to-date published literature.
L264: rewrite as "...naringinase was added..."
L273: what "TBDPSCl" stands for?
L278: part of the sentence lacks.
L512: "S.E.M."?
L516: "an" instead of "the".
L520: delete "the"s before "various", "flavonol", "flavonols" and "organic".
L534-750: the list contains too many references owing excessive self-citation. I highly respect authors' previous scientific contribution, but this confronts the scientific ethics.
Reviewer 3 Report
About publication this manuscript is decided some points which should be improved and complemented by the Authors.
Page 2 (Lines 54-56): Did the authors isolate helichrysoside from Helichrysum kraussii or did they only make use of the information contained in the literature? Please clarify it. Page 3 (Line 80): The highest content of 17 in the reaction mixture was observed after 1.5 h but not after 2h. Page 8 (Line 229): Please provide information: How long did it take to mix a suspension of rutin and was a water bath used for this? Page 7 (Line 209): Please enter the full TBDMSCl shortcut name. Page 260 (Line 260): Where in the manuscript is the formula for compound 22a? Page 8 (252): If the authors describe the synthesis of helichrysoside they should present it graphically.Author Response
We are grateful to your reviewing our manuscript and providing valuable suggestions to improve the manuscript. We have incorporated all your comments and suggestions in our revised manuscript. I hope this new manuscript is acceptable for publication in Int. J. Mol. Sci.
Reviewer #3
About publication this manuscript is decided some points which should be improved and complemented by the Authors.
Page 2 (Lines 54-56): Did the authors isolate helichrysoside from Helichrysum kraussiior did they only make use of the information contained in the literature? Please clarify it.→
Thank you for the suggestion. We used synthesized helichrysoside in the study. The indicated sentences have been revised as following.
(Page 2 Line 54-58)
This paper deals with practical synthesis and glucose tolerance-improving activity of helichrysoside (1 = quercetin 3-O-(6''-O-trans-p-coumaroyl)-b-D-glucopyranoside), isolated from Helichrysum kraussii and H. stoechas [24, 25] by other research groups. Furthermore, synthetic study of the related analogs of 1 (2–15, Figure 1) were also carried out as well as characterization of its structural requirements for glucose and lipid metabolism in HepG2 cells.
Page 3 (Line 80): The highest content of 17 in the reaction mixture was observed after 1.5 h but not after 2h.
→
Thank you for the valuable point out. There was no considerable difference the conversion from 19 to 17 between 1 to 2 h after the start of the reaction. However, since a relatively large amount of unreacted substance 19 remained after 1.5 h, the reaction was performed for 2 h.
Page 8 (Line 229): Please provide information: How long did it take to mix a suspension of rutin and was a water bath used for this?
→
Thank you for the suggestion. The indicated sentences have been revised as following.
(Page 8 Line 236-237)
A suspension of rutin (19, 100.0 mg) in H2O (50 mL) was mixed and stirred at 50 °C under water bath in a few minutes. Then naringinase (5.0 mg) was added to the suspension to start the reaction.
Page 7 (Line 209): Please enter the full TBDMSCl shortcut name. Page 260 (Line 260): Where in the manuscript is the formula for compound 22a? Page 8 (252): If the authors describe the synthesis of helichrysoside they should present it graphically.
→
According to the comment, the full name of TBDPSCl was added in Page 7 Line 216. In addition, we inserted Scheme 1 in Page 3 Lines 92–98.
Round 2
Reviewer 1 Report
The authors completely ignored the most scientifically relevant comments from the previous round, especially those related to the concept of the study, the existence of more efficient methods for the synthesis of isoquercitrin, the yields and especially purity (isoquercitrin is practically non-separable from quercetin!) of the compounds synthesized and tested in cell lines, the selection of compounds for the two test on cell cultures. All these are in the first review by this reviewer. Only replies to the reviewer 2 comments can be found in the revised manuscript, but authors mention some of my comments as well (changes in the title and the suggestion to move a table into the supplement). Therefore, they for sure have read the review and simply intentionally omitted the most problematic comments.
Moreover, the OGGT test used as the main in vivo test of the title compound is known to lead to false positive result. Any compound administered together with glucose in this test will delay glucose absorption and therefore influence its plasma level.
Author Response
We are grateful to your reviewing our manuscript and providing valuable suggestions to improve the manuscript. We have incorporated all your comments and suggestions in our revised manuscript. I hope this new manuscript is acceptable for publication in Int. J. Mol. Sci.
Reviewer #1
The authors completely ignored the most scientifically relevant comments from the previous round, especially those related to the concept of the study, the existence of more efficient methods for the synthesis of isoquercitrin, the yields and especially purity (isoquercitrin is practically non-separable from quercetin!) of the compounds synthesized and tested in cell lines, the selection of compounds for the two test on cell cultures. All these are in the first review by this reviewer. Only replies to the reviewer 2 comments can be found in the revised manuscript, but authors mention some of my comments as well (changes in the title and the suggestion to move a table into the supplement). Therefore, they for sure have read the review and simply intentionally omitted the most problematic comments.→
This is generally recognized widely that flavonoid glycosides (such as isoquercitrin) and the aglycons (quercetin) are easy to separate using an open silica gel column.
And, there are no reports on the synthesis of helichrysoside in such a large scale and in a short step than in this manuscript. The most emphasized is the practical method. Both reaction substrate rutin and the enzyme naringinase is the most inexpensive and commercially available reagent. As for the concept of this study, we have modified both Abstract and Introduction section according to the comments from both Reviewer #1 and #2.
Moreover, the OGGT test used as the main in vivo test of the title compound is known to lead to false positive result. Any compound administered together with glucose in this test will delay glucose absorption and therefore influence its plasma level.
→
As for the GTT test, we have described that a significant reduction was observed in AUC of blood glucose level up to 2 h after glucose loading. In addition, it is clear from Fig. 2, there is no delay in glucose absorption. The biggest misunderstanding in our study, glucose was loaded intraperitoneally (i.p.), not orally, described in 3. Materials and Methods section. Therefore, the sample administered orally and glucose did not interact, and the result of helichrysoside is reliable.
Reviewer 2 Report
I find the authors addressed most of the concerns raised in the previous review round. I suggest acceptance.
Author Response
We are grateful to your reviewing our manuscript and providing valuable suggestions to improve the manuscript. We have incorporated all your comments and suggestions in our revised manuscript. I hope this new manuscript is acceptable for publication in Int. J. Mol. Sci.
Reviewer #2
I find the authors addressed most of the concerns raised in the previous review round. I suggest acceptance.
→
Thank you very much for your time and kind consideration.